# Constrained MLAMBDA Method for Multi-GNSS Structural Health Monitoring

**DOI:** 10.3390/s19204462

**Published:** 2019-10-15

**Authors:** Haiyang Li, Guigen Nie, Dezhong Chen, Shuguang Wu, Kezhi Wang

**Affiliations:** 1GNSS Research Center, Wuhan University, Wuhan 430079, China; 2Southern Surveying and Mapping Geographic Company, Guangzhou 510663, China

**Keywords:** GNSS deformation monitoring, integer ambiguity resolution, multi-GNSS, MLAMBDA, known priori information, constraint conditions

## Abstract

Deformation monitoring of engineering structures using the advanced Global Navigation Satellite System (GNSS) has attracted research interest due to its high-precision, constant availability and global coverage. However, GNSS application requires precise coordinates of points of interest through quick and reliable resolution of integer ambiguities in carrier phase measurements. Conventional integer ambiguity resolution algorithms have been extensively researched indeed in the past few decades, although the application of GNSS to structural health monitoring is still limited. In particular, known a priori information related to the structure of a body of interest is not normally considered. This study proposes a composite strategy that incorporates modified least-squares ambiguity decorrelation adjustment (MLAMBDA) method with priori information of the structural deformation. Data from the observation sites of Baishazhou Bridge are used to test method performance. Compared to MLAMBDA methods that do not consider priori information, the ambiguity success rate (ASR) improves by 20% for global navigation satellite system (GLONASS) and 10% for Multi-GNSS, while running time is reduced by 60 s for a single system and 180 s for Multi-GNSS system. Experimental results of Teaching Experiment Building indicate that our constrained MLAMBDA method improves positioning accuracy and meets the requirements of structural health monitoring, suggesting that the proposed strategy presents an improved integer ambiguity resolution algorithm.

## 1. Introduction

GNSS is one of the most efficient tools in structural health monitoring due to its high precision, continuity and real-time performance, irrespective of weather conditions [1]. For this reason, many studies have been conducted into GNSS applications. Li proposed a real-time interference monitoring technique for GNSS based on twin support vectors which used twin support vector machine (TWSVM) to satisfy the requirement in real-time interference monitoring [2]. Barzaghi analyzed the GNSS-estimated displacements on the Eleonora D’Arborea (Cantoniera) dam and compared it with pendulum data. In his study, the results led to the conclusion that GNSS technique can be applied to dam monitoring allowing adenser description, both in space and time, of the dam displacements than the one based on pendulum observations [3]. Meng proposed a new system for large bridge monitoring, referred to as GNSS structural health monitoring (GeoSHM), which was crucial to ensure the safety, serviceability, and sustainability of large infrastructures, the preliminary results in his study showed GeoSHM Demo Project had huge potential of the state-of-the-art Earth Observation in offering a better understanding of ground movements surrounding bridge sites [4]. Integer ambiguity resolution is required to obtain precise real-time coordinates of monitoring site. The ambiguity search region is like an ellipsoid and the search radius is the flag of its size [5]. The LAMBDA method is widely used to fix ambiguities, which involves sorted-sequence double-difference (DD) measurements [6]. Decorrelation is used to reduce the correlation of DD ambiguities and sequence sorting is employed to reorder DD ambiguities according to their variance. A modified LAMBDA method is proposed by Chang to update search radius every time [7]. Park proposed the strategy to reduce the search radius of ambiguity through the constraints of observation accuracy and variation between adjacent epochs, and Wang employed constrained LAMBDA for Global Positioning System (GPS) attitude determination by further deriving Park’s algorithm. In Park and Wang’s study, the integer ambiguity resolution for GPS performed better than unconstrained algorithm [8,9]. Landry R combined the recursive least squares (RLS) method and LAMBDA to improve the baseline estimation and attitude accuracy [10]. Gong reduced search radius using the baseline information. In his study, the prior information of baseline length, heading, and pitch were all integrated into objective function [11]. However, an ambiguity resolution algorithm for use in GNSS structural health monitoring has not yet been developed, and this deficiency limits fast and efficient integer ambiguity resolution in this field.

The development of GNSS, together with the decreasing cost of hardware and increasing reliability of high-frequency observations, promotes structural monitoring applications. GPS automatic monitoring of the Geheyan dam, China, achieved positioning accuracy of 0.5 mm in the horizontal direction and 1.0 mm for vertical direction [12]. The wind and structural health monitoring system (WASHMS) for the three long-span cable-supported bridges in Hong Kong provides a GPS-monitoring observation accuracy of 10–20 mm, correctly identifying a structural vibration frequency of 10 Hz [13,14,15]. Previous studies have been largely based on GPS, and as a result, the majority of integer ambiguity resolution algorithms have been developed for use with GPS. The development of Multi-GNSS strengthens the geometric observation structure with an increased number of satellites, allowing for positioning-algorithm optimization.

Previous studies have proved that priori information obtained at the preparation stage of GNSS structural health monitoring is key to algorithms optimization. The similar single-difference model provided an approach to calculating the monitoring site position that involves constructing vector models of satellites and stations [16]. This method uses single-difference equations for stations, and requires receiver clock errors obtained from priori information. Its mathematical model restricts the observation range of structural monitoring because it skips the ambiguity-fixing process. Based on the fact that a point of interest changes only slightly between adjacent epochs, Bai and Ren proposed a single-epoch algorithm with irrespective of cycle by increasing the sampling rate [17]. Algorithms employing real-time kinematics provide templates for converting priori information to integer ambiguity resolution constraints. Algorithms based on ambiguity resolution with constraint equation (ARCE) enable the fixing of ambiguities with single measurements, even in case of long baselines [18].

This study proposes a strategy that utilizes geometric model of satellite-station distance and deformation characteristics to obtain constraints for the MLAMBDA method. The large error in pseudo-range observations means that it is not advisable to make constraints based on these. Structural health monitoring involves ultra-short baseline observation, and baseline conversion is suitable for obtaining the necessary constraints. In contrast to Park’s method, the standard observation accuracy is replaced by coordinate-based and environmental errors. Environmental errors have a great impact on the progress of the coordinate solution, while the maximum deformation value places additional constraints on each satellite system for stricter constraints. The intersection of the above two constraints is the final constraints for the MLAMBDA method in GNSS structural health monitoring.

## 2. Materials and Methods

As opposed to employing ordinary relative positions, deformation monitoring solves the baseline and requires pre-observations to obtain priori information. Priori information provides valid constraints for structural health monitoring. Reducing the ambiguity search radius through the use of converted constraints greatly reduces the running time of the proposed algorithm.

For the Multi-GNSS algorithm, the burden of ambiguity searching will abruptly increase with an increasing number of satellites, and a large variance value could adversely affect the ambiguity resolution procedure. The Multi-GNSS algorithm provides more observations for calculating the monitoring coordinates. In the case for which a single satellite system is poorly observed, its data can be corrected using other systems. Hence, determining suitable MLAMBDA method constraints for use in GNSS structural monitoring is key to optimizing the proposed algorithm. This study proposes a strategy aimed at constraints for the MLAMBDA method in GNSS structural monitoring. The progress of the proposed approach is detailed in Figure 1.

As shown in the flowchart in Figure 1, the detailed steps of the proposed algorithm are as follows:Conduct LD decomposition of ambiguity variance matrix $Q=L^{T}DL$, where Q is the ambiguity variance matrix and D is a diagonal matrix.Construct integer ambiguity transform matrix Z using matrix L: $Q=L^{T}DL=Z^{T}Q_{1}{}^{T}Q_{1}Z$.Transfer the float solution into a fixed solution $z=Z^{T}a$,where a is the DD integer ambiguity. The constraints are also transformed to $zn=Z^{T}b$, where b denotes the constraints.Search for the optimal integer ambiguity. The MLAMBDA method involves shrinking the search radius through updating the radius as $\sum \frac{\left(z_{j}-\bar{z_{j}}\right)}{d_{j}}$, where j is the serial number of the fixed ambiguity in matrix z.Transfer the search result of DD ambiguities back to the integer ambiguities.

### 2.1. Constraint of Deformation Information

The MLAMBDA method can be divided into two parts: multidimensional integer transformation for decorrelation and searching. The formula of ambiguity searching is expressed as follows.
(1)$$f=\sum_{i=1}^{n}\frac{\left(z_{i}-\bar{z_{i}}\right)}{d}\leq r^{2}$$
where d denotes the variance matrix after integer transformation.; $z_{i}$ is the float solution for the ambiguity; $\bar{z_{i}}$ is the alternative integer solution for the ambiguity; and r is the radius that will be updated at every epoch. As shown in Equation (1), the size of the search radius is directly related to the searching time of the integer ambiguity resolution. The search radius will be large for high variance values. In this situation, searching ambiguities might prove to be a difficult task. Moreover, in the case of Multi-GNSS monitoring, there are more satellites. These factors could lead to difficulties if we adopt the MLAMBDA method without constraints. The derivation of constraints is discussed in the following section.

Priori deformation information can be divided into two parts: monitoring-sites characteristics and the baseline information. Constraints with structural characteristics are used to reduce the search radius based on environmental and coordinate accuracy. The linear GNSS phase observation equation is represented in the following form.
(2)∇ΔΦ=A×a+λ×∇ΔN+∇Δε
where ∇ΔΦ is the DD carrier phase observation matrix; a is the deflection matrix of the site of interest; A is the coefficients matrix of the DD observation equation; *λ* is the carrierphase length; ∇Δ*N* is the DD integer ambiguity matrix; and ∇Δε is the DD residual correction matrix. The ambiguity formula can be derived from the following equation.
(3)$$\sigma_{\nabla \Delta N}=\frac{\sqrt{\sigma_{\left(\nabla \Delta \Phi -A\times a\right)}{}^{2}+\sigma_{\nabla \Delta \varepsilon }{}^{2}}}{\lambda }=\frac{\sqrt{\sigma_{a}{}^{2}+\sigma_{\nabla \Delta \varepsilon }{}^{2}}}{\lambda }$$
where $\sigma_{\nabla \Delta \Phi }$ is accuracy of carrier phase which is negligible with respect to $\sigma_{a}$; $\sigma_{a}$ is the standard deviation(STD) matrix for the monitoring site; and $\sigma_{\nabla \Delta \varepsilon }$ is the STD matrix for the environmental error. The coordinate error can be calculated by the priori observation. The first constraint of the DD integer ambiguity can be described in the following form.
(4)$$\nabla \Delta N_{0}-\delta_{\nabla \Delta N}\leq b_{1}\leq \nabla \Delta N_{0}+\delta_{\nabla \Delta N}$$
where $\nabla \Delta N_{0}$ is the float solution; $b_{1}$ is the first constraint for all the satellites; $\delta_{\nabla \Delta N}=\frac{\beta \times \sigma_{\nabla \Delta N}}{\lambda }$; β is the double-tail quantile value of the t (Student) distribution, where the confidence level is 1 − *α* and the freedom is *f*, where *f* is the number of satellite and *α* = 0.01. In general, *β* = 3 is assumed to be the value for the rule of thumb [19].

Based on the fact that the displacement of an observer between adjacent epochs is smaller than the maximum deformation values, an additional constraint is used to obtain stricter constraints. In contrast to the approach proposed by Park and Dai, the constraints of the maximum deformation values are set as the additional constraints. After the procedure of integer Z transformation, DD ambiguities are sorted according to their variance. From Equation (1), we see that DD ambiguities with larger variances have larger search radius. For this season, it is particularly important to select suitable satellites for the additional constraints. Information from the integer Z transformation is used as a reference for this selection process. If there are insufficient observations, these insufficient observations are grouped to obtain further restrictions. In this way, there are more restrictions than in the case of Dai’s algorithm. The deflection between two adjacent epochs must be less than the maximum deformation value. This relationship is expressed as Equation (5).
(5)$$d^{2}\geq \Delta E^{2}+\Delta N^{2}+\Delta U^{2}$$
where *d* is the max deformation value of the structures; E,N,U are the variations in the local coordinate system. In contrast to the work of Dai, a Kalman filter is used to make the prediction for the next epoch [20]. Based on this prediction, the maximum value is substituted by subtracting the observation value of the previous epoch from the predicted value.
(6)$$d^{2}={\left(\nabla \Delta \varepsilon -\lambda \times \nabla \Delta N\right)}^{T}\times {\left(A^{-1}\right)}^{T}\times A^{-1}\times (\nabla \Delta \varepsilon -\lambda \times \nabla \Delta N)={\left(\nabla \Delta \varepsilon -\lambda \times \nabla \Delta N\right)}^{T}\times {\left(A\times A^{T}\right)}^{-1}\times (\nabla \Delta \varepsilon -\lambda \times \nabla \Delta N)$$

Cholesky decomposition of is to transform the positive definite matrix $AA^{T}$ to a upper triangle matrix and a lower triangle matrix. And it is expressed as $AA^{T}=LL^{T}$, where L is the lower matrix, $L^{-1}=\left[\begin{array}{ccc} l_{11} & & \\ l_{21} & l_{22} & \\ l_{31} & l_{32} & l_{33} \end{array}\right]$ and $l_{ij}$ is the element in row i and column j of L. Equation (6) then reflects the matrix of deformation. Based on this matrix, the partial constraints $b_{2}$ of the first selected satellite are expressed as follows.
(7)$$MAX(-\delta_{\nabla \Delta N_{1}},-\frac{d}{\lambda \times l_{11}}+\frac{\nabla \Delta \varepsilon_{1}}{\lambda })\leq b_{2}\leq MIN(\delta_{\nabla \Delta N_{1}},\frac{d}{\lambda \times l_{11}}+\frac{\nabla \Delta \varepsilon_{1}}{\lambda })$$

According to the process employed for the first selected satellite, the partial constraints of the second and third satellite are determined as follows.
(8)$$\left\{ \begin{array}{l} MAX(-\delta_{\nabla \Delta N_{2}},\gamma_{1})\leq b_{3}\leq MIN(\delta_{\nabla \Delta N_{2}},\gamma_{2})\\ MAX(-\delta_{\nabla \Delta N_{3}},\gamma_{3})\leq b_{4}\leq MIN(\delta_{\nabla \Delta N_{3}},\gamma_{4}) \end{array}\right.$$
where $\gamma_{1}=\frac{-\sqrt{d^{2}-B^{2}}+l_{21}(\nabla \Delta \varepsilon_{1}-\lambda \times \nabla \Delta N_{1})+l_{22}\times \nabla \Delta \varepsilon_{2}}{\lambda \times l_{22}}$ and $\gamma_{2}$, $\gamma_{3}$, $\gamma_{4}$ have the same form, which is derived from the Equation (6). These three additional constraints allow for further restrictions on Multi-GNSS structural health monitoring. After this module, the ambiguity constraints are further reduced.

In previous studies, the part of the residual ∇Δ*ε* in Equation (6) is an ellipsis but this is integral to guaranteeing the mathematical rationality of the above procedures. The observation environment of structural health monitoring means that the influence of ∇Δ*ε* is remarkable. Another reason to retain the residual ∇Δ*ε* is that when selecting the satellites, the Multi-GNSS corrections, such as inter system bias (ISB), differential code bias (DCB), or fractional cycle bias (FCB), are assigned to the residual ∇Δ*ε*. The process may be interrupted if the residual is ignored.

The use of constraints based on baseline length is another effective method for optimizing algorithms. Kores proposed precise gravity recovery and climate experiment (GRACE) baseline determination using GPS [21]. In his study, the resulting solution matches the GRACE K-Band ranging system measurements with an accuracy of 1 mm, whereby 83% of the DD ambiguities are resolved. We propose a strategy in which satellite-station distances are obtained using a mathematical model. The satellite-station distance is transformed by the baseline length and satellite-station vector, expressed by A in Equation (2). A mathematical conversion allows us to describe the system in the following form.
(9)$$-round(\frac{\left|\Delta A\right|\times m}{\lambda }+0.5)<b_{5}<round(\frac{\left|\Delta A\right|\times m}{\lambda }+0.5)$$
where $b_{5}$ is the another constraints for all the satellites; |ΔA| is the length of ΔA; and m is the length of the baseline which is updated in every epoch. There are many ways to obtain a priori baseline distance, such as using laser rangefinders. The algorithmic conditions in this study are independent of satellite systems. The constraints of all the satellites and frequencies can be obtained from the baseline length using known priori information. To ease the computational burden on the system, this constraint is placed on the position values for the reference and monitoring stations. The precise coordinates of the two observation stations are among the priori data, which could be used to optimize positioning progress. These are all the constraints employed in this study, using the priori information of monitoring-site characteristics and baseline information.

### 2.2. Constraints on Multi-GNSS MLAMBDA

In this section, we elaborate on our approach to determining intersection and its transformation into constraints for used in the proposed MLAMBDA method. The selection of constraints for deformation characteristics using satellites is based on the order of matrix Z. The constraint of the satellite-station distance is in the original order. The final constraint on the above two constraints is their intersection. The constraint on Multi-GNSS MLAMBDA is expressed in the Equation (10).
(10)$$Zn=Z^{T}\times b$$
where Zn is the final constraint on the Multi-GNSS and b is the intersection of $b_{1},b_{2},b_{3},b_{4},b_{5}$. Z is the integer transform matrix for the MLAMBDA method $Zn=Z^{T}\times \nabla \Delta N$. The integer transform matrix Z for Multi-GNSS MLAMBDA is different from that of a single system. Matrix block operation is considered in the procedure. Progress is sorted according to the system and frequency. The order of matrix Z is recorded for use as an additional constraint on the maximum deformation value. This section makes the constraints for the Multi-GNSS MLAMBDA method.

## 3. Experiments and Analysis

Experiments on both the Baishazhou Bridge, Wuhan, China, and the Teaching Experiment building of Wuhan University, China, were performed to test the advantages and feasibility of the proposed constrained MLAMBDA method, respectively. The running environment for the program was vs2012 C language on a 64 bit Windows 8.1 personal computer, with 4GB RAM and an Intel(R) Core (TM) i5-6200U processor CPU @2.30 GHz 2.40 GHz.

### 3.1. Experiment on the Baishazhou Bridge

The advantages of the proposed constrained MLAMBDA method are evaluated through an experiment conducted on the Baishazhou Bridge. The running time of the program, ambiguity success rate (ASR), ambiguity alternative group size and epoch-to-first fixed ambiguity were selected as the evaluation indicators. The experiment was conducted for 1 h (GPST 12:00:07–13:00:08, September 26th, 2016) on 10 Hz GPS/beidou navigation satellite system (BDS)/GLONASS data of the Baishazhou bridge. Baishazhou Bridge is a cable-stayed bridge comprised of double towers and double cable planes. The site of interest sites is referred to as S012 in the middle span. Observations were performed using a ComNav-K508 system. The monitoring coordinate accuracy of this experiment was 0.1 m, 0.1 m, 0.3 m. The priori observation also provided the precise coordinates for the site of interest.

Table 1 shows a comparison of the experimental results for monitoring site S012. The proposed constrained MLAMBDA method provided a 60 s reduction in running time for both GPS and BDS, a 90 s reduction for GLONASS, a 120 s reduction for GPS+BDS, and a 180 s reduction for Multi-GNSS. With the respect to the epoch-to-first fixed ambiguity, the proposed method optimized the initialization process for ambiguities, which fixes the ambiguity in the first epoch for BDS, while initialization for GLONASS is poor in this experiment. BDS and GPS both outperform GLONASS and the initialization of Multi-GNSS initialization is better than GLONASS. 

In the case of the ASR, the proposed constrained MLAMBDA method provides an improvement of 4 % for GPS, 0.2% for BDS, 24% for GLONASS, 7% for GPS+BDS, and 10% for Multi-GNSS. GPS and BDS outperform GLONASS, wherein the BDS observation is particularly accurate. Optimization is the most significant in the case of GLONASS due to its poor observation. These results indicate that the proposed constrained MLAMBDA method optimizes ambiguity resolution and that the observation of GLONASS is relatively poor. Low running time and higher ASR are important to make accurate calculation in real-time structural healthy monitoring.

Figure 2 shows the results of the ambiguity alternative group size for three single systems and Multi-GNSS. The ambiguity alternative group size reflects the radius of the search radius in single-epoch resolution. When using the proposed algorithm, the size of GPS, BDS and Multi-GNSS ambiguity alternative groups were each reduced by approximately four, whereas the GLONASS ambiguity alternative groups was reduced by approximately six. The changes in satellite numbers in Figure 3 highlight the abrupt fluctuations of each of the four figures shown in Figure 2. All four results exhibited significant fluctuations at approximately the 12000 epoch. Figure 2 indicates that the proposed constrained MLAMBDA method provided a high level of restriction in contrast to the unconstrained method.

Figure 3 shows the change in the satellite numbers during the experiment on the S012 site. There were a sufficient number of single-system satellites during the observation. The numbers of satellites for the three single-system were approximately equal before the 25,000th epoch, after which there were three fewer satellites for GLONASS. The number of satellite for Multi-GNSS fluctuated at approximately the 12,000th epoch.

Table 2 shows the accuracy statistics for the S012 site. The proposed constrained MLAMBDA method outperforms the unconstrained approach in terms of observation accuracy, to varying degrees. GPS and BDS exhibit less improvement than others because of the high accuracy of their raw data. GLONASS shows the most significant improvement, because of the poor quality of its raw data. From the overall results, we may conclude that the conclusion that the constrained MLAMBDA outperforms the unconstrained method.

### 3.2. Experiment on the Teaching Experiment Building of Wuhan University

The experiment on Baishazhou Bridge indicated that the constrained MLAMBDA method presents advantages such as a high fixed ambiguity success rate and low running time, compared to the unconstrained approach. An experiment on the Teaching Experiment building of Wuhan University was performed to verify the feasibility of the constrained MLAMBDA method in structural monitoring. The experiment was conducted for 90 min (GPST 07:00:00–08:50:00, January 30th, 2018) by using 1 Hz GPS + GLONASS data. In this experiment, a directional mobile observation base was used to verify the accuracy of the algorithm in terms of structural monitoring. The maximum moving ranges of the mobile observation base are 12 cm (E direction), 17 cm (N direction) and 0 cm (U direction). A 5-min-long observation is conducted as a group, and the base was moved by 1 cm in the E and U directions for each time. Internal accuracy and external accuracy factors were used to evaluate positioning accuracy.

Table 3 shows the results for the Teaching Experiment building experiment. Because of the high-quality experimental environment and small impact of errors, this test is suitable for verifying the feasibility of using the constrained MLAMBDA method for structural monitoring. The program running time is reduced by approximately 20 s compared to the unconstrained algorithm, and the ASR is increased by 6%. Since a small dataset is used in this experiment, the improvement in running time is not significant. Similar to the conclusions drawn from Table 1, the constrained algorithm effectively improves the GNSS positioning algorithm in structural health monitoring.

Figure 4 shows the results of the mobile quantitative observation experiment. We can see that in the E and U directions, the calculation results are in good agreement with the actual displacement, indicating that the proposed constrained algorithm can accurately fix ambiguities in the deformation monitoring experiment. The deviation between the calculated result and the actual displacement is different in two directions. This experiment moves in the E and N directions, and the settlement result in the N direction is better than that in the E direction. Though better than the characteristics of GNSS observation, the deviation in the U direction is larger than those in the N and E directions. This deviation is most pronounced for the epochs from 0–300 s. Due to limitations in the initial coordinate accuracy of the monitoring station prior information, the residuals are relatively obvious at the beginning of this period. Figure 4 shows that using the constrained MLAMBDA method for structural monitoring can meet the requirements of centimeter-level structural monitoring.

Table 4 shows the accuracy analysis results for the experiment on the Teaching Experiment building. In addition to the three directions of E, N and U, there is a precision statistic for the total displacement V. From Table 4 we can see that the constrained algorithm effectively improves the internal and external accuracy of the observation points for the E, N, and U directions and V. When applied to structural monitoring, the constrained MLAMBDA method can reach an accuracy of 3 mm in the horizontal direction and 5mm in the elevation direction. However, the accuracy of the observation points in the three directions E, N and U also significantly differ. The accuracy in the E direction is slightly worse than that in the N direction, and is the worst in the U direction.

## 4. Conclusions

We propose a composite optimization strategy for an MLAMBDA-based GNSS structural health monitoring to reduce the running time and improve ASR. Drawing on the work of Park and Dai, priori information were used to transform into effective constraints for the MLAMBDA method. The necessary priori information included baseline length, impact on environmental errors, and the maximum deformation values of structures. A Kalman filter was used to obtain predictions for the next epoch and prediction substitutes for the max deformation values. In addition, this approach provides next-epoch prediction substitutes for the maximum deformation values. As a result, the proposed constrained MLAMBDA method improves on the ASR by approximately 20% and reduces the running time of Multi-GNSS by 180 s. Experiments conducted on the Teaching Experiment building of Wuhan University indicate that the proposed algorithm can achieve positioning accuracy of 3 mm in the horizontal direction and 5 mm in the elevation direction. The proposed algorithm improves the positioning accuracy of the monitoring points and fully meets the requirements of structural monitoring.

However, in the process of formula derivation, the proposed algorithm is still subject to certain limitations. In particular, the satellite selection is aimed at adding more constraints to the search radius of ambiguity. We also note that the residuals are placed into an equation that need more rigorous studies so it can be refined. Further optimization of the float solution and refinement error are required to obtain stricter constraints.

## Figures and Tables

**Figure 1 sensors-19-04462-f001:**
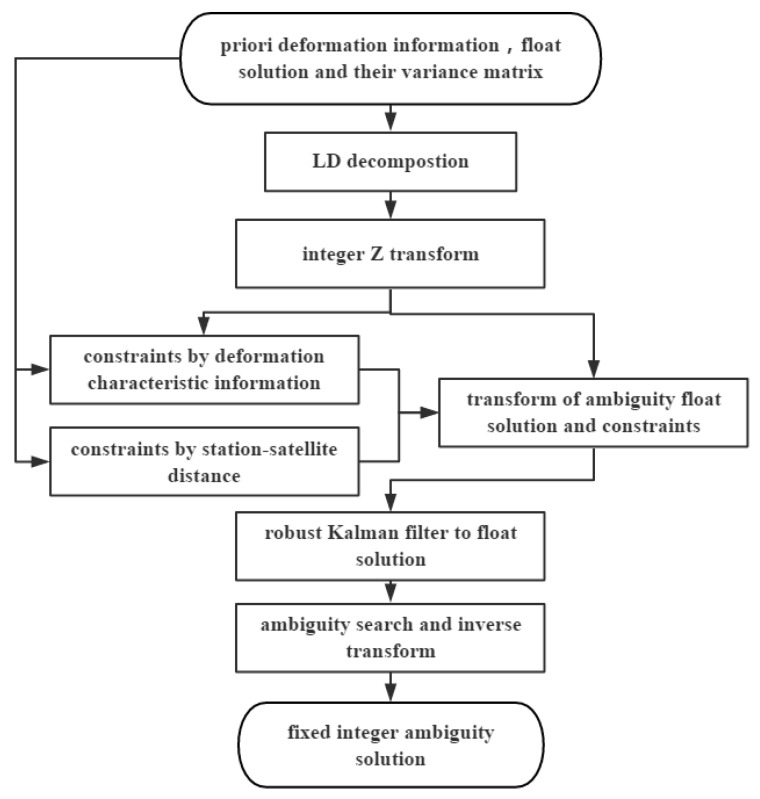
Flowchart for the proposed constrained Multi-GNSS MLAMBDA Method.

**Figure 2 sensors-19-04462-f002:**
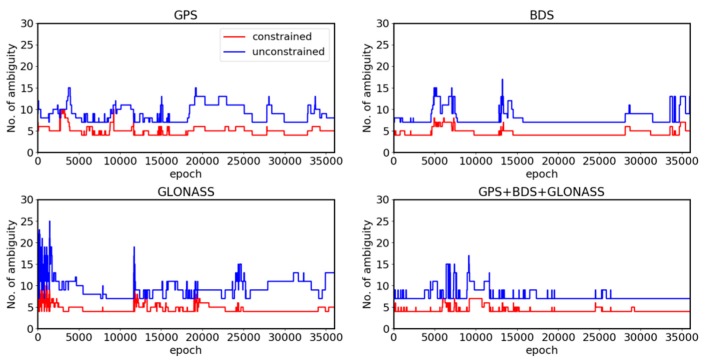
Ambiguity alternative group size for the S012 monitoring site.

**Figure 3 sensors-19-04462-f003:**
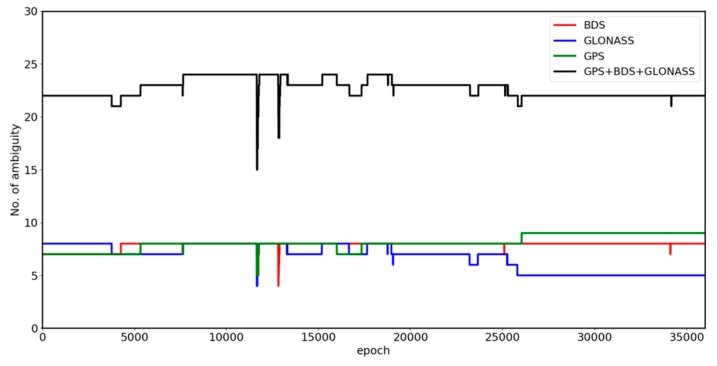
Satellite number of the S012 experiment.

**Figure 4 sensors-19-04462-f004:**
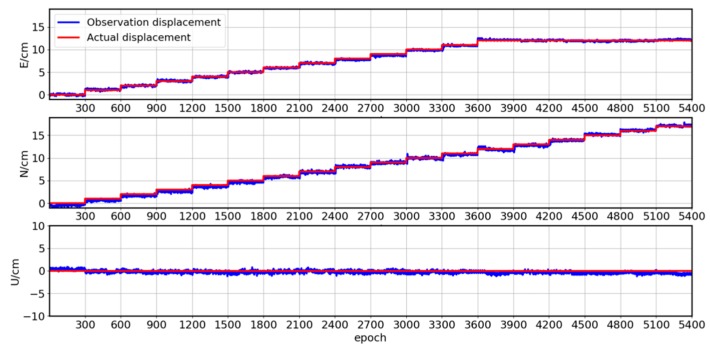
Displacements of mobile quantitative observations in the E, N, and U directions for the proposed constrained algorithm.

**Table 1 sensors-19-04462-t001:** Comparison of experimental results for the site of interest S012.

Baseline Length (m)	Multi-GNSS	Method	Running Time (s)	Epoch-to-First Fixed Ambiguity	ASR (%)
2480.4751	GPS	constrained	719.451	5	95.9
		unconstrained	788.561	60	91.6
	BDS	constrained	768.111	1	99.8
		unconstrained	838.462	51	99.6
	GLONASS	constrained	674.581	2231	89.2
		unconstrained	760.271	5272	65.1
	GPS+BDS	constrained	1162.402	1	94.0
		unconstrained	1280.622	1	87.6
	GPS+BDS+GLONASS	constrained	1763.982	90	76.9
		unconstrained	1945.271	1655	66.3

**Table 2 sensors-19-04462-t002:** Internal accuracy statistics of the S012 monitoring site for the E, N, U and total displacement V directions (in mm).

Multi-GNSS	Method	E	N	U	V
GPS	constrained	6.13	7.84	13.56	16.82
	unconstrained	9.51	11.03	18.47	23.52
BDS	constrained	3.29	3.87	7.15	8.77
	unconstrained	4.13	4.24	8.64	10.47
GLONASS	constrained	11.84	13.57	24.52	35.35
	unconstrained	176.42	195.52	403.12	481.52
GPS+BDS	constrained	7.84	8.61	15.45	19.35
	unconstrained	13.46	14.57	26.87	33.40
GPS+BDS+GLONASS	constrained	72.14	85.21	143.87	182.11
	unconstrained	154.58	162.82	334.87	403.17

**Table 3 sensors-19-04462-t003:** Comparison of experimental results for the Teaching Experiment building, Wuhan.

Baseline Length (m)	Method	Running Time (s)	ASR (%)
427.8865	constrained	486.681	99.8
	unconstrained	508.970	93.8

**Table 4 sensors-19-04462-t004:** Accuracy statistics for tests on the Teaching Experiment building for the E, N and U directions, and the total displacement V (in mm).

		E	N	U	V
Constrained	Internal accuracy	2.65	2.14	4.31	5.49
	External accuracy	3.89	3.22	5.05	7.14
Unconstrained	Internal accuracy	3.05	2.69	5.13	6.53
	External accuracy	3.98	3.54	6.11	8.11

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
