# Peer review of "Constrained MLAMBDA Method for Multi-GNSS Structural Health Monitoring"

_sensors, 2019, doi:10.3390/s19204462_

Round 1

Reviewer 1 Report

I have attached annotated version of pdf file. I think this is valuable work but some elements can be improved.

Looking at the results, it seems that GLONASS is performing far worse then other systems. What is the reason for not computing GPS+BDS solution?

Some drawings with experiments design (baseline lengths, receivers placement) would be useful. Also Survey condition (eg DOP factors, numer of satellites) would give some idea about the experiment.

Author Response

Thanks for your comment and suggestions for my manuscript. The follows are my explanations.

1.Looking at the results, it seems that GLONASS is performing far worse then other systems. What is the reason for not computing GPS+BDS solution?

As suggested, we added results for the GPS+BDS solution, see l.228 l.235 and Table 1. Because of their high-quality raw data,the improvement is not significant.

2.Some drawings with experiments design (baseline lengths, receivers placement) would be useful. Also Survey condition (eg DOP factors, numer of satellites) would give some idea about the experiment.

Details on the baseline lengths and related experimental details are provided on l.180. we use a mathematical model to make constraints as formula(9)

3.I have attached annotated version of pdf file. I think this is valuable work but some elements can be improved.

Thank you for your suggestions in pdf. I have altered the manuscript as you suggestions and upload a new manuscript. I added parameter explanation of t distribution in l.143, explanation of matrix L in l.161, c language in l.210, deletion of the explanation of round(x) and adjust the format.

Reviewer 2 Report

This paper tackles structural monitoring carried on with GPS receiver. To estimate the distance (baseline) between two points of a structure a CDGPS technique has been used. Fundamental for minimizing the error in baseline determination is exploiting the integer nature of the GPS ambiguity. To estimate the integer part of the ambiguity a constrained MLAMBDA method has been used.

Usually lambda method requires a validation of the ambiguity to be fixed. In this paper the ambiguity fixing seems to be performed before the integer transformation (eq. 10). Indeed, only the ambiguities fulfilling the constraints in section 2.1 are transformed to integer. It is not clear if after an ambiguity is fixed it is kept constant in the following epochs.

The paper is not clearly written and a thorough revision of the English must be performed. In the following there are some issues to be fixed:

Introduction

Line 29/44 – Introduction must describe better and more clearly the approaches used in literature. In addition some concept must be defined before used. E.g. define what is the search radius before using it in line 41.

Line 36 – “sequence to double-difference(DD) carrier phase measurements” could be misleading. I would replace with "sorted sequence of DD measurements".

Line 41 -  write the acronym for RLS

Line 64/66 – Besides using triple frequency receiver, there are other ways for solving ambiguity problems over long baseline, i.e. [1]. However, I don’t understand why you cite long baseline problems, if your research is related to small baselines.

Line 72 – It is not clear how the satellite-station distance it is estimated.

Method

The methodology is very difficult to understand, and a thorough revision must be performed. The author should provide a more organized description of all the part of the used method. This section must be rewritten, in order to help the reader to understand. Here there are some suggestions:  

Figure 1 – you cannot let a figure explain the model you’re using. The flowchart depicted in the figure must be detailed and widely described in the manuscript for clarifying the algorithm described by the paper.  

Line 117 – replace “of the magnitude difference to” with “is negligible with respect to”

Line 121 – what is ∇ΔN0?

Line 122 – what do you mean for “t distribution”? You may mean τ (tau) distribution.

There is a confusion in the usage of ∇ΔNx in equation (4),(7)-(9). Sometimes (∇ΔN1 ∇ΔN6) it is referred to be a constraint in the other cases it is the ambiguity of the first, second or third satellites. It is not clear how the satellites are ordered and why are not considered the ambiguities of satellites greater than three. As far as the satellite sorting is concerned, liens 128-129 clarify that after the integer transformation the satellites are sorted by their variance, but the integer transformation (equation (10)) occurs after the satellites’ ambiguities selection, as written in line 178-179.

Line 143 - there is not ∇ΔN2 in equation (7).

Line (159) – Which mathematical model do you use to estimate the satellite-station distance? They are basically pseudoranges. It is strange that you use an a-priori super precise estimate of the solution if your aim is to obtain the solution itself.

Line 161 - there is not A in equation (8).

Result

Figure 4 – the discussion of the figure should be performed in the text not in the figure caption

Result should show also the success rate on the estimated ambiguities, comparing the estimated ambiguities with the true ones.

[1]          Kroes, R., Montenbruck, O., Bertiger, W., and Visser, P., “Precise GRACE baseline determination using GPS,” GPS Solutions, vol. 9, 2005, pp. 21–31

Author Response

1.The paper is not clearly written and a thorough revision of the English must be performed.

The Introduction and Method have been revised to improve readability and comprehension. The English language issues throughout the text have been corrected though extensive revision.

2.Line 29/44 – Introduction must describe better and more clearly the approaches used in literature. In addition some concept must be defined before used. E.g. define what is the search radius before using it in line 41.

From l.29, the Introduction has been revised to provide more details on the other approaches in the literature.

3.Line 36 – “sequence to double-difference(DD) carrier phase measurements” could be misleading. I would replace with "sorted sequence of DD measurements".

The term "sorted-sequence DD measurements" has been adopted, see l.45.

4.Line 41 -  write the acronym for RLS

The acronym definition has been added for RLS, see l.52.

5.Line 64/66 – Besides using triple frequency receiver, there are other ways for solving ambiguity problems over long baseline, i.e. [1]. However, I don’t understand why you cite long baseline problems, if your research is related to small baselines.

References to long-baseline issues have been removed.

6.Line 72 – It is not clear how the satellite-station distance it is estimated.

Details are provided on satellite-station measurements, see l.176. and formula(9) is the estimation of the satellite-station distance.

7.Figure 1 – you cannot let a figure explain the model you’re using. The flowchart depicted in the figure must be detailed and widely described in the manuscript for clarifying the algorithm described by the paper.  

A detailed description of the flow chart in Figure 1 is now included in the text, see l.106.

8.Line 117 – replace “of the magnitude difference to” with “is negligible with respect to”

The phrase "negligible with respect to" has been adopted, see l.137.

9.Line 121 – what is ∇ΔN0?

I definition of is provided on l.141.

10.Line 122 – what do you mean for “t distribution”? You may mean τ (tau) distribution.

I did mean T(Student) distribution, this has been corrected in l.142.

11.There is a confusion in the usage of ∇ΔNx in equation (4),(7)-(9). Sometimes (∇ΔN1 ∇ΔN6) it is referred to be a constraint in the other cases it is the ambiguity of the first, second or third satellites. It is not clear how the satellites are ordered and why are not considered the ambiguities of satellites greater than three. As far as the satellite sorting is concerned, liens 128-129 clarify that after the integer transformation the satellites are sorted by their variance, but the integer transformation (equation (10)) occurs after the satellites’ ambiguities selection, as written in line 178-179.

I have corrected the symbols in the formula in the new manuscript. I'm sorry to have trouble to you. It's my carelessness to the fault.

12.Line 143 - there is not ∇ΔN2 in equation (7).

The new symbol replaced ∇ΔN in l.162

13.Line (159) – Which mathematical model do you use to estimate the satellite-station distance? They are basically pseudoranges. It is strange that you use an a-priori super precise estimate of the solution if your aim is to obtain the solution itself.

An expanded description of our method for obtaining the satellite-station distance has been included in l.176 and formula(9). this mathematical model is a rough but fast estimation and it is useful to MLAMBDA.

14.Line 161 - there is not A in equation (8).

It is my carelessness and A is in equation(2) in l.182,and I have corrected it.

15.Figure 4 – the discussion of the figure should be performed in the text not in the figure caption

Result should show also the success rate on the estimated ambiguities, comparing the estimated ambiguities with the true ones.

This is a format mistake and I have upload my new manuscript to correct it.

16.[1] Kroes, R., Montenbruck, O., Bertiger, W., and Visser, P., “Precise GRACE baseline determination using GPS,” GPS Solutions, vol. 9, 2005, pp. 21–31

Thank you for your recommendation. I have read the paper and his study is aimed at the station-satellite distance. I used a mathematical model in formula (9) to make rough but fast estimation and I will make a deep learning on your recommendation later. 

Reviewer 3 Report

Dear Authors,

in your paper you propose a modified LAMBDA algorithm aiming at enhancing the Ambiguity Success Rate and considering multi-GNSSsignals. Despite you propose the algorithm in the framework of structural monitoring, it is not clear the advancement in this sense. In such field of application, improvements in the positioning accuracy are much more important than a quick fixing of the ambiguities. Your results tell about faster fixing of few minutes but on a 24h/365gg monitoring this is relatively important. However, based on the results of the second described experiment, there is an accuracy improvement and, in this case, the title of your manuscript finally fits. In my opinion, if you want to maintain the title as it is, you should focus on this aspect. As an alternative, just clarify why you think that a speed-up in solving the phase ambiguities would help in monitoring a structure.

After my general impressions on the subject, let me point out that the style and formatting of the manuscript require deep refinements.

There are lots of missing spaces, truncated sentences, grammar errors, missing explanations of the symbols appearing in the equations, wrong references to figures, not proper reference formatting. Table 3 should be quite simple to be read  but, as it is now, the format requires to be changed.

In my opinion, the paper is not ready to be accepted in the present form and also the content requires some refinement.

I hope that you will find my suggestions helpful for the next steps.

Best reagards.

Author Response

1.in your paper you propose a modified LAMBDA algorithm aiming at enhancing the Ambiguity Success Rate and considering multi-GNSS signals. Despite you propose the algorithm in the framework of structural monitoring, it is not clear the advancement in this sense. In such field of application, improvements in the positioning accuracy are much more important than a quick fixing of the ambiguities. Your results tell about faster fixing of few minutes but on a 24h/365gg monitoring this is relatively important. However, based on the results of the second described experiment, there is an accuracy improvement and, in this case, the title of your manuscript finally fits. In my opinion, if you want to maintain the title as it is, you should focus on this aspect. As an alternative, just clarify why you think that a speed-up in solving the phase ambiguities would help in monitoring a structure.

Thank you for your suggestions. My work is aimed at single epoch structural health monitoring algorithm and the experiments are used to verify the advantages. Low running time and higher ASR are very important to optimize in real-time deformation monitoring. The first experiment is to test its advantages and the second advantage is to verify its feasibility. Low running time is important in real-time monitoring and it is the key to make early warning.

2.After my general impressions on the subject, let me point out that the style and formatting of the manuscript require deep refinements.

The Introduction and Method have been revised to improve readability and comprehension. The English language issues throughout the text have been corrected though extensive revision. And the formatting of the manuscript has been corrected in the new upload manuscript.

3.There are lots of missing spaces, truncated sentences, grammar errors, missing explanations of the symbols appearing in the equations, wrong references to figures, not proper reference formatting. Table 3 should be quite simple to be read  but, as it is now, the format requires to be changed.

I have corrected the formatting errors as the journal request and revised the content in the new upload manuscript. I'm sorry to have trouble to you.

Round 2

Reviewer 1 Report

The authors responded to all my comments and the manuscript is significantly improved.

Author Response

Thank you for your suggestions and it helped me improve the manuscript.

Best wishes to you.

Reviewer 2 Report

The authors have made significant changes to improve the quality of the manuscript. By adding Cholesky decomposition and matrix L, the constraint on the first three satellites resulted more clear. However the components of matrix L in line 161 must be clarified. 

All the discussion about the constrains seems now more clear. Some English world spelling must be revised, e.g. line 145 i believe it is observer not observatory.

Author Response

Thank you for your suggestions and it helped me improved the manuscript. I have corrected the error as your suggestions.

1.The authors have made significant changes to improve the quality of the manuscript. By adding Cholesky decomposition and matrix L, the constraint on the first three satellites resulted more clear. However the components of matrix L in line 161 must be clarified. 

As your suggested,adding explanation of cholesky decomposition and matrix L and making explanation to the components of matrix L.

2.All the discussion about the constrains seems now more clear. Some English world spelling must be revised, e.g. line 145 i believe it is observer not observatory.

I'm sorry to trouble you and it was my carelessness. I have corrected it in the new manuscript.

Reviewer 3 Report

Dear Authors,

style and formatting are definitely improved with respect the previous version. There are still some grammar errors but much less than before.

About the content, I still have doubts about the research goal. In my opinion the research is fine but not suitable in the framework of structural monitoring. However, I prefer leaving the final decision to the Editorial board.

Just a oiunt, have a look at Table 3, I suppose a typing error is present in the GLONASS-constrained-V line.

Author Response

Thank you for your suggestions and it helped me improve the manuscript. The error in table 3 has been corrected and it is my carelessness. As for the first suggestion, maybe it can be expressed as follows: the program is not available in practical structural health monitoring. This study is only aimed at improving the process of integer ambiguity resolution in GNSS structural health monitoring. I will make further study to update the algorithm in future.

Best wishes